# A Bioluminescence-Based Ex Vivo Burn Wound Model for Real-Time Assessment of Novel Phage-Inspired Enzybiotics

**DOI:** 10.3390/pharmaceutics14122553

**Published:** 2022-11-22

**Authors:** Vincent De Maesschalck, Diana Gutiérrez, Jan Paeshuyse, Yves Briers, Greetje Vande Velde, Rob Lavigne

**Affiliations:** 1Department of Biosystems, KU Leuven, Kasteelpark Arenberg 21, 3001 Leuven, Belgium; 2Department of Biotechnology, Ghent University, Valentin Vaerwyckweg 1, 9000 Gent, Belgium; 3Biomedical MRI, Department of Imaging and Pathology, KU Leuven, Herestraat 49, 3000 Leuven, Belgium

**Keywords:** bioluminescence, antimicrobial activity screening, ex vivo model, lysin

## Abstract

The silent pandemic of antibiotic resistance is thriving, prompting the urgent need for the development of new antibacterial drugs. However, within the preclinical pipeline, in vitro screening conditions can differ significantly from the final in vivo settings. To bridge the gap between in vitro and in vivo assays, we developed a pig-skin-based bioluminescent ex vivo burn wound infection model, enabling real-time assessment of antibacterials in a longitudinal, non-destructive manner. We provide a proof-of-concept for *A. baumannii* NCTC13423, a multidrug-resistant clinical isolate, which was equipped with the *luxCDABE* operon as a reporter using a Tn7-based tagging system. This bioluminescence model provided a linear correlation between the number of bacteria and a broad dynamic range (10^4^ to 10^9^ CFU). This longitudinal model was subsequently validated using a fast-acting enzybiotic, 1D10. Since this model combines a realistic, clinically relevant yet strictly controlled environment with real-time measurement of bacterial burden, we put forward this ex vivo model as a valuable tool to assess the preclinical potential of novel phage-inspired enzybiotics.

## 1. Introduction

Due to the current antimicrobial resistance crisis, we have arrived in a post-antibiotic era in which previously easy-to-treat infectious diseases can no longer be treated efficiently [1]. Consequently, a renewed interest in the development of new antibacterials is emerging, following alarming reports published by several leading public institutions [1,2,3,4,5]. While the current clinical pipeline primarily comprises chemical derivatives of the currently existing (small-molecule) antibiotics, more diverse strategies are entering the preclinical pipeline, including the development of vaccines, virulence-attenuating antibodies, and the use of bacteriophages and (engineered) lysins [4,6,7]. These new antibacterial strategies act in a completely distinct manner from standard-of-care antibiotics discovered in once successful approaches, such as the Waksman platform [8]. Moreover, since the introduction of small-molecule antibiotics, several insights in microbiology shed a different light on wound infections. For instance, pathogens involved in (chronic) wound infections tend to reside in biofilms, and many infections tend to be polymicrobial [9,10]. Therefore, new disease models for bacterial infections may provide new insights in the performance of new antibacterial approaches. Several wound models have been developed to mimic infection by these (often drug-resistant) opportunistic pathogens and, hence, test the activity of newly developed antibacterials [11]. Nonetheless, the vast majority of these models are either in vitro or in vivo models [12]. Ex vivo models could provide an appropriate middle ground, combining a more realistic matrix with a tightly controlled environment. To mimic wound infections, a pig skin matrix is highly relevant from a translational viewpoint, as it shares striking similarities with human skin [13]. Therefore, several studies included a porcine-skin-based approach to study microbial growth in more realistic settings [14,15,16,17,18]. Notably, almost all these studies mention a destructive approach relying on metabolic activity, qPCR, or cell counts. By contrast, bioluminescence-based models enable longitudinal data collection, limit the number of explants needed for similar experiments and facilitate a more reliable/robust statistical comparison [17].

Once discovered as a natural phenomenon [19], bioluminescence imaging (BLI) is a widely used approach when studying bacterial burden in vivo [20,21]. Several different reporter systems based on bioluminescence have been developed, relying on a luciferase and an exogenously added substrate [22,23,24]. However, this step is bypassed when using the bacterial luciferase system—grouped together in the *luxCDABE* operon, all required genes are present to provide a luminescent signal [25]. With the various genome-editing tools currently available, genetic insertion of a set of genes has become increasingly efficient, resulting in a stable copy of this operon into the genome of a desired host. While BLI was already widely used in animal studies to visualize bacterial load [20,26], this approach recently gained more attention in the microbiology field to replace cumbersome and time-consuming replica plating [17,27].

In the search for an easy-to-implement realistic wound model, we developed a real-time, imaging-compatible ex vivo pig skin model to close the gap between in vitro and in vivo models. By generating a flexible plasmid collection with different antibiotic selection markers and the *luxCDABE* operon, we tagged a clinical *A. baumannii* isolate and established proof-of-concept for high-throughput real-time antibacterial efficiency screening with a recently published enzybiotic, 1D10. This engineered lysin was selected for retaining its activity in serum and its ability to reduce the bacterial count in an ex vivo wound model [28].

## 2. Materials and Methods

### 2.1. Bacterial Strains, Growth, and Media

For molecular cloning, *Escherichia coli* PIR2 (Thermo Fischer Scientific, Waltham, MA, USA) was used, as this strain is compatible with the R6K origin of replication of pBG13, pBGlux, and pBGlux_CmR. For the three-parental mating, *E. coli* HB101 harboring pRK2013 was used [29]. *Acinetobacter baumannii* NCTC13423 was selected as a clinical isolate to be used in the ex vivo model [30,31]. All cultures were grown in Lysogeny Broth (LB) and incubated overnight (37 °C, 150 rpm) or on solid LB agar (37 °C), supplemented with the necessary selectable markers (Appendix A).

### 2.2. Construction of Versatile Tagging Plasmids, pBGlux and pBGlux_CmR

Starting from the pBG13 backbone, new plasmids, pBGlux and pBGlux_CmR, were created (Figure 1). Therefore, the *aacC1* gene conferring resistance to gentamycin was exchanged by a chloramphenicol acetyltransferase coding sequence using Type IIs restriction enzyme-based cloning. The latter gene was Phusion-PCR amplified from pLemo isolated from *E. coli* Lemo(DE3) cells (New England Biolabs, Ipswich, MA, USA) using a set of primers annealing directly in front of and behind the ORF of this gene (primers used: CmR_F and CmR_R, available in Appendix A); T_a_ = 61.5 °C, t_e_ = 30s). The tails of these primers were designed comprising a *Bsa*I restriction site followed by the appropriate overhang, allowing in-frame assembly of this gene. Next, an inverse PCR was performed on pBG13, creating complementary overhangs upon *Bsa*I restriction (primers used: pBG13_Abmark_F and pBG13_Abmark_R, available in Appendix A; T_a_ = 61.5 °C, t_e_ = 130 s). After PCR purification (GeneJet PCR purification kit, Thermo Fisher Scientific, Waltham, MA, USA), both PCR products were digested and ligated using *Bsa*I and T4 DNA ligase, respectively (Thermo Fisher Scientific). Next, chemical transformation using the Rubidium Chloride method of the newly generated plasmid to *E. coli* PIR2 and subsequent plating on selective LB agar comprising 30 µg/mL chloramphenicol allowed for selection of appropriate clones [32].

To replace the existing *msfGFP* ORF with the *luxCDABE* operon, a Gibson assembly was used [33]. For this, the required Phusion–PCR reactions were performed to generate the appropriate homologous sequences (T_a_ = 63.2 °C, t_e_ = 200 s for the *luxCDABE* operon with primers GA_luxCDABE_F and GA_luxCDABE_R; T_a_ = 60.5 °C, t_e_ = 130 s for the pBG13/pBG13_CmR backbone with primers GA_pBG_F and GA_pBG_R (Appendix A)). Next, PCR amplification products were purified and a Gibson assembly reaction was performed using an equimolar amount of insert and backbone. Finally, 8 µL of the reaction mix was transformed to *E. coli* PIR2 and plated on selective LB agar. Clones emitting light were selected for DNA sequencing of the entire plasmid (MiniSeq, Illumina, San Diego, CA, USA). Library preparation was performed using an Illumina DNA prep kit (Illumina, USA) with a starting concentration of 5 ng/µL of DNA. Plasmids were pooled, and each plasmid accounted for 0.2% of the total DNA pool. This pool was denatured following the MiniSeq denaturing protocol A and sequenced with a MiniSeq Mid outpout 300 cycles reagent kit (Illumina, USA).

### 2.3. Tagging A Clinical Isolate with the Newly Created Tagging Plasmid, pBGlux_CmR

*A. baumannii* NCTC13423 was tagged with pBGlux_CmR using three-parental mating [31]. Briefly, 100 µL of an overnight culture of *A. baumannii* NCTC 13423, *E. coli* HB101 containing pRK2013 and *E. coli* PIR2 containing pBG13_CmR were pooled and pelleted, washed with 500 µL LB, and resuspended in 50 µL LB. This mixture was spotted on LB agar and incubated overnight at 37 °C. Upon collection of the cells on the plate with a 0.85% (*w*/*v*) NaCl solution, the mixture was plated on LB agar containing 90 µg/mL chloramphenicol and 30 µg/mL gentamicin for counter selection. Clones emitting light were verified using PCR to verify the correct insertion in the genome of *A. baumannii* (T_a_ = 50.3 °C, t_e_ = 40 s, with primer pair AB_glmS_v2_F and Tn7_R (Appendix A)).

### 2.4. Preparation of Pig Skin Explants for Use in the Ex Vivo Model

Pig skin from euthanized pigs was obtained from TRANSfarm (ECD n° P040/2020), shaved, and disinfected with 70% ethanol. A stainless-steel rod with a diameter of 4.8 mm was heated in a flame for 30 s. Each explant was burned for 1 s with this rod and disinfected with desinfectol. Next, the explants were placed in a black 24-well plate (Eppendorf, Hamburg, Germany), on top of 1 mL sterile, physiological agar (0.9% NaCl, 0.7% bacteriological agar, pH 5.5).

### 2.5. Establishing A Correlation between the RLU on the Explant and the CFUs Recovered from the Explant

Explants were prepared as described above, inoculated with 10^5^ CFUs of *A. baumannii* NCTC13423::*luxCDABE*, and incubated statically at 30 °C. After 0 h, 1 h, 3 h, 5 h, 7 h, 9 h, 12 h, and 24 h, the luminescence of the explants was determined on a Clariostar Plus (10 s integration time, normalized for 1 s, focal height 20 mm, enhanced dynamic range; BMG Labtech, Ortenberg, Germany). Next, three explants were taken for each time point and homogenized using a sterile homogenizer tube containing 1.5 mL PBS (Precellys Evolution, 7200 rpm, 2 × 45” with 10” interval time; Bertin Corp., Rockville, MD, USA). Next, a tenfold dilution series was made in 200 µL PBS, and luminescence was measured (Clariostar Plus, 10 s integration time, normalized for 1 s, focal height: 13 mm, enhanced dynamic range). The baseline was measured from wells containing 200 µL PBS (n = 9). Subsequently, 100 µL of the desired dilutions were plated on LB agar and incubated overnight (30 °C). Bacterial counts were determined by counting luminescent colonies on these plates. All experiments were performed in triplicate.

### 2.6. Testing Antibacterial Activity in the Bioluminescent Ex Vivo Model

The pig skin explants were prepared as described above and inoculated with 10^5^ CFUs of *A. baumannii* NCTC13423::*luxCDABE*. Next, the pig skin explants were incubated statically at 30 °C in the Clariostar Plus. Every five minutes, luminescence was measured (10 s integration time, normalized for 1 s, focal height 20 mm, enhanced dynamic range). After five hours, the desired dose of the antibacterial engineered lysin 1D10 was sprayed onto the pig skin explants, as performed in [28]. All experiments were performed in triplicate.

### 2.7. Data Analysis

Data was analyzed and graphically illustrated using Microsoft Excel and JMP Pro 16 (SAS institute, Cary, NC, USA).

## 3. Results

### 3.1. Establishing A Flexible Plasmid Collection to Tag Non-Model Organisms

To create a widely applicable tagging system for bacteria, a plasmid compatible with the Tn7 system, as described by Choi et al., was constructed [34] (Figure 1). Briefly, this system enables site-specific integration into the bacterial chromosome using Tn7 attachment (attTn7) sites located downstream of the *glmS* gene. These sites have been found to be present in over twenty bacterial species, mostly Gram-negative, including opportunistic pathogens such as *Pseudomonas aeruginosa*, *A. baumannii*, and *Salmonella enterica*. Therefore, the SEVA sibling, pBG13, was used as a backbone, enabling constitutive expression of a reporter gene [35]. Using Type IIs-based cloning based on *Bsa*I, the open reading frame containing the *aacC1* gene conferring resistance to gentamycin was exchanged for a chloramphenicol acetyltransferase gene. Next, we introduced the *luxCDABE* from *Photorhabdus luminescens* as a reporter system. As a template for the *luxCDABE* operon, pSEVA426 was used [36]. More specifically, the operon was incorporated into the backbone using a Gibson assembly, resulting in pBGlux and pBGlux_CmR, respectively (Figure 1) [33]. These plasmids were sequence-verified using Illumina sequencing.

The applicability range of this system was illustrated by tagging a non-model strain with this newly developed suicide plasmid. One of the most urgent multidrug-resistant threats in hospital environments is carbapenem-resistant *A. baumannii* [1]. Therefore, *A. baumannii* NCTC13423, a multidrug-resistant and clinically relevant isolate [31], was tagged with pBGlux_CmR, as screening revealed this strain to be susceptible to 60 µg/mL chloramphenicol. Tagging was achieved using the three-parental mating method. The selection of clones harboring the correct insertion was based on both phenotypic and genotypic assessment, selecting light-emitting clones, and verifying genomic insertion by PCR. Next, the growth rate and the capacity to form biofilms of the newly created *A. baumannii NCTC13423*::luxCDABE was compared with an untagged *A. baumannii* NCTC13423. These assays confirm that the tagged strain is suitable for use in the model, as differences in growth rate and biofilm formation were minimal (Appendix A). Moreover, the emission spectrum of this strain was determined and corresponded to previously reported emission spectra of *luxCDABE* (Appendix A) [37], indicating that the reporter strain was appropriate for further use.

### 3.2. Integration of the Bioluminescent Reporter Strain in the Ex Vivo Pig Skin Model

As a next step, the tagged strain was utilized in the ex vivo pig skin model. In this model, pig skin explants comprising a circular burn wound were inoculated with 10^5^ bacteria. Growth of this strain in the model was monitored over a time span of 24 h. Furthermore, after each time point (0 h, 1 h, 3 h, 5 h, 7 h, 9 h, 12 h, and 24 h), the explants were processed, and both the bioluminescence intensity of the homogenates and the number of bacteria on each explant were determined (Figure 2).

At each time point starting from inoculation of the explant with 10^5^ CFU until the end of the experiment after 24 h, the measured bioluminescence signal of the homogenates was significantly higher than that of the baseline (Student’s *t*-test assuming unequal variances, Appendix A). This indicates that the bioluminescent approach is sufficiently sensitive in identifying the establishment of a bacterial infection on the pig skin. Moreover, since a bacterial count of 10^5^ CFU is often used to clinically diagnose infection [38], one can conclude that this model provided the bacteria with a suitable environment to proliferate to levels similar to those in patients diagnosed with a state of infection, indicating its potential clinical relevance. Another interesting feature of using luminescence (compared with replica plating) is its dynamic range—where the same wells could be used for luminescence read-out, a dilution series had to be made to obtain countable plates. Plating the correct dilution requires either preliminary experiments or radically increasing the number of plates. Moreover, this BLI approach enables monitoring bacterial growth on a single explant as a function of time, whereas a CFU-based approach would require a separate explant for each individual time point to be examined.

Next, the logarithmic transformation of the luminescence intensity obtained directly from the explants (RLU/s/explant) and the logarithmic transformation of the amount of colony forming units per piece of pig skin (CFU/explant) were plotted to evaluate their agreement (Figure 3). A linear correlation was obtained spanning five orders of magnitude, resulting in a dynamic range ranging between 10^4^ and 10^9^ bacteria, with an R^2^ value of 0.9543. As a result, such a calibration curve now allows us to use bioluminescence intensity measurements obtained from the explant as an alternative read-out for bacterial burden, with log transformations of light output from the explant and the number of bacteria on the explant showing a linear correlation.

### 3.3. Proof-of-Concept with A Bacteriophage-Derived Engineered Lysin, 1D10

The eventual goal of this ex vivo model is to efficiently assess the antibacterial activity of newly developed antibacterial components in more realistic wound settings. A novel class of promising antibiotics are bacteriophage-derived lysins and their engineered variants [4,39,40]. These enzymes rapidly degrade the peptidoglycan layer of their specific target bacteria resulting in cell lysis. These enzyme-based antibiotics or enzybiotics are currently in the (pre-)clinical development stage [41,42] (ClinicalTrials.gov, NCT04160468). Here, we used the recently reported engineered lysin 1D10, which has a minimal inhibitory concentration (MIC) of 6 µg/mL against *A. baumannii* NCTC13423 and performed well in an ex vivo wound model [28]. After infecting the explants with 10^5^ CFU, luminescence intensity was monitored over time. Five hours post infection, three explants were treated with either buffer, 25, 50, or 100 µg 1D10. An instant drop in luminescence was observed for the treated explants, characteristic of the mode-of-action of (engineered) lysins (Figure 4). The longitudinal BLI approach allowed us to observe that, within 45 min after this treatment (Table 1), the bioluminescence intensity increased again due to regrowth of treatment-surviving cells. However, the measured luminescence intensity was shown to be lower for the treated explants compared with the untreated ones, as was described previously for this compound when relying on CFU counts [28].

The longitudinal nature of luminescence intensity measurements allows us to determine the dynamics of bacterial growth and antibacterial treatment and to define a relevant time frame for further analysis of antibacterial efficacy. In this case, we focused on the time frame between 3 h and 7 h post-infection to assess the antibacterial effect in more detail (Appendix A). Therefore, we normalized the data to their own baseline values, i.e., by dividing the measured values by the bioluminescence intensity measured for each explant right before addition of the engineered lysin. This yielded a drop in (relative) luminescence counts of almost 80% in RLU/s/explant when explants were treated with 50 µg or more of the engineered lysin, displayed in Table 1 and Appendix A.

The system was disturbed when the plate had to be extracted from the luminometer for the application of the engineered lysin, which is reflected in the small drop in bioluminescence intensity for the untreated control (Appendix A, Table 1). Compared with this control, all other treatments showed a significant drop in (relative) bioluminescence intensity at significance level α = 0.05 (Table 1). No significant difference could be observed among treatments. Due to the high time resolution of this approach, it was possible to discriminate among the different treatments. Not only is a lower drop visible for treatment with 25 µg 1D10, the antibacterial effect is shorter compared with the other treatments. This is illustrated in Table 1: the minimal relative bioluminescence intensity is obtained after 0.278 h (16 min) for a treatment with 25 µg 1D10, whereas the minima for treatments with 50 µg and 100 µg 1D10 were obtained at least 38 min after treatment. This antibacterial effect is significantly longer compared with the control treatment at α = 0.05. From this analysis, a significant difference could be found between the untreated control and the three treatments (α = 0.05) when comparing the area under the curve in the first four hours after treatment.

A final approach to compare the different concentrations of engineered lysin that were tested relied on the fitting of an exponential curve between 5 h and 9 h post-infection (Figure 5, Table 2). The derivative of this fitted curve was calculated representing the growth rate the bacteria on the skin for each treatment condition. Then, we could calculate the growth rate over this exponential growth period as indicated in Table 2. This mathematical analysis confirmed the visual representation (Figure 5), indicating a decreased growth rate within the four hours after treatment with the enzybiotic. More specifically, we observed a difference of one order of magnitude in growth rate, expressed in RLU s^−1^ h^−1^, between the samples treated with 50 µg or higher in comparison with the untreated control. As a result, all calculated parameters suggest using a dose of at least 50 µg of this engineered compound, which was also the dose that was used in previous ex vivo studies with this compound [28].

## 4. Discussion

Ex vivo models hold the potential to bridge the gap between easy-to-perform assays and realistic, but labor-intensive and ethically burdened, in vivo experiments. Here, we describe a longitudinal, bioluminescence-based burn wound model for which we use skin from porcine origin, maximizing its translational potential [13]. We illustrated the applicability of this model with an in-house engineered lysin, calculating several parameters confirming an antibacterial effect and giving insights into the dynamics of antibacterial treatment.

By expanding the SEVA repository with a broad-spectrum, standardized, Tn7-based tagging vector, either clinical isolates or non-model organisms can potentially be included in this wound model. While the currently designed SEVA siblings allow selection for gentamycin and chloramphenicol, this marker can straightforwardly be exchanged using type IIs-based restriction. Theoretically, this system allows insertion in most bacterial species as the integrative attP sites are present in virtually all bacteria. However, only Gram-negative bacteria have been reported to be site-specifically tagged with this system [27,34,43]. This system can be used in eukaryotes as well as for *Saccharomyces cerevisiae*, which enables the expansion of the model to fungal infections [44]. However, in this case, Tn7 insertion is observed to be random rather than downstream from *glmS*. This highlights the need to compare the fitness of the tagged strain with its wild type for relevant phenotypic characteristics. Nonetheless, it is also recommended to verify the latter when tagging Gram-negative bacteria, even though this type of insertion is believed to be neutral [34].

Furthermore, a bioluminescent, longitudinal approach has several major advantages over commonly used destructive approaches, such as replica plating. A longitudinal approach not only drastically reduces hands-on time, time-to-results, and the number of explants needed to carry out an experiment, it also allows us to follow the same piece of skin throughout time in a continuous measurement, reducing (biological) variation between different time points during statistical analysis (Figure 3). In addition, sterility of the explant is not as strict a requirement when using a bioluminescent reporter strain.

Interestingly, bioluminescence intensity measurements also cover a larger dynamic range compared with bacterial counts. While a dilution series of the homogenized explant is required to plate the correct dilution, the luminometer spans the entire range of emitted light from 10^4^ to 10^9^ bacteria. Moreover, this approach enables an immediate view on the dynamics of the newly developed antibacterial, which also allows us to assess the influence of a different dose of antibacterial (Table 1 and Table 2, Figure 5 and Appendix A). We observed a shorter antibacterial effect for the explants treated with 25 µg of engineered lysin in comparison with higher doses of this component, which was not reported previously, possibly due to the lower time resolution. These higher doses resulted in a significantly longer antibacterial effect in comparison with the untreated control. Insights in antibacterial dynamics of a novel antibacterial compound could, in theory, also be gained by using a destructive approach but would need prior knowledge of a previous experiment to determine a “time frame of interest” and is, hence, more labor-intensive. However, obtaining a similar time-resolution as with BLI throughout the entire experiment is practically impossible, further highlighting the significant advantages of bioluminescence measurements over destructive methods.

As the majority of (burn) wound infections tend to be polymicrobial with several opportunistic pathogens colonizing specific niches in the wound environment [9,10], this model could be adapted to follow two or more bacterial species at the same time. As several engineered luciferase variants, including a red-shifted variant [45], have been developed, this model could be upgraded to a more complex, multispecies variant. Moreover, it should be noticed that this model should not be limited to bacteria only: several pathogenic fungi have been tagged recently with luciferase expressing genes [23,24,46,47]. To conclude, options to tailor or expand this model are plentiful, edging us closer to realistic settings.

## Figures and Tables

**Figure 1 pharmaceutics-14-02553-f001:**
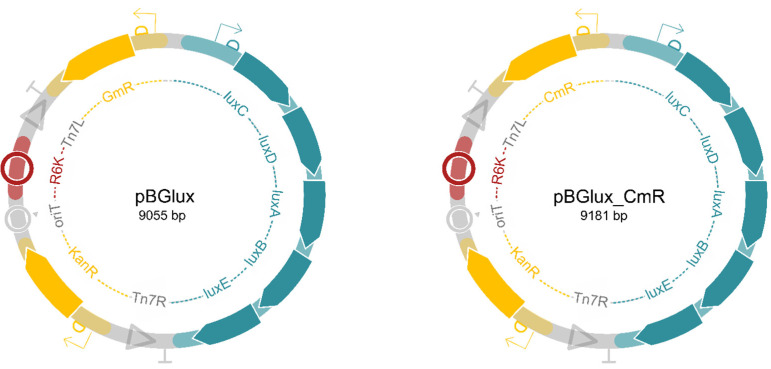
**Schematic overview of the newly created SEVA-siblings, pBGlux and pBGlux_CmR**. Both plasmids encode Tn7 sites, enabling efficient transposition. The R6K origin of replication is a suicide origin avoiding the plasmid to exist as a plasmid in the host. Eventually, the cargo flanked by Tn7L and Tn7R will be inserted into the host genome, including the *luxCDABE* operon and an antibiotic selectable marker. Here, we included a gentamycin resistance gene (pBGlux) and a chloramphenicol resistance gene (pBGlux_CmR), but these can easily be exchanged using Type IIs-based cloning, depending on the antibiotic resistance profile of the host. These plasmids are available upon request.

**Figure 2 pharmaceutics-14-02553-f002:**
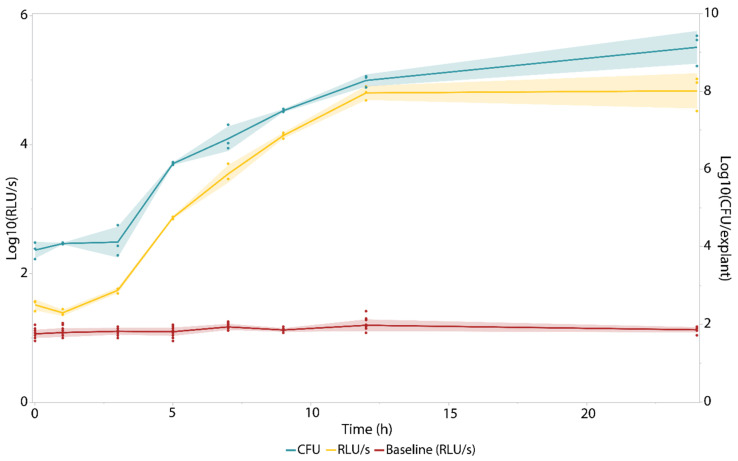
**Growth of *A. baumannii* NCTC13423::*luxCDABE* in the ex vivo pig skin burn wound model.** Bacteria were recovered from the skin using a homogenizer, and a tenfold dilution series was made to determine the amount of colony forming units per explant (colony forming units per explant (CFU/explant), right *y*-axis marked in teal). Then, the bioluminescence intensity of this homogenate was measured (expressed in relative light units per second (relative light units per second (RLU/s)), left *y*-axis marked in yellow). The baseline (in red) indicates the detection limit of the bioluminescence measurements. Each time point consists of three biological replicates (n = 3), whereas the sample size of the baseline comprised nine replicates (n = 9) The error bands indicate one standard deviation from the mean.

**Figure 3 pharmaceutics-14-02553-f003:**
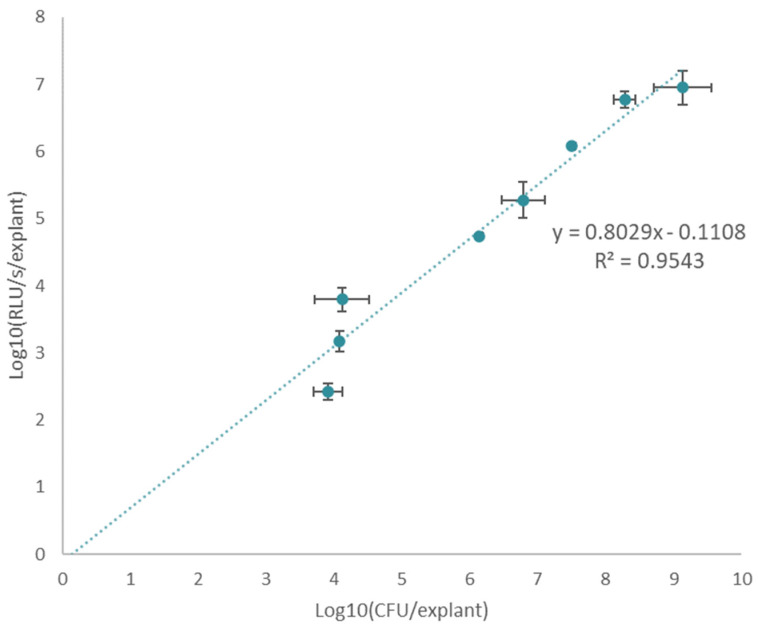
**The luminescent signal is proportional to the CFU count recovered from the explant, enabling an efficient measurement for bacterial burden.** A linear relation between the log transformations of both luminescent signal (RLU/s/explant) obtained from the explant and bacterial number (CFU/explant) was obtained, with a R^2^ of 0.9543. This confirms the linear relationship between bioluminescence obtained from the explant and bacterial burden over a period of 24 h after inoculation of the explant. Each time point consists of three biological replicates (n = 3). The error bars represent one standard deviation from the mean.

**Figure 4 pharmaceutics-14-02553-f004:**
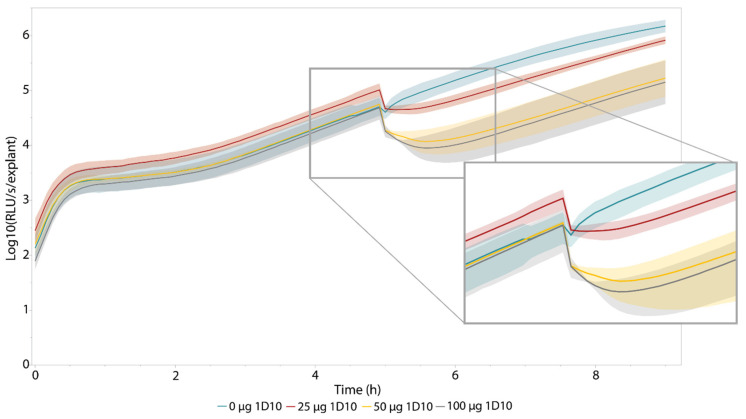
**Longitudinal bioluminescence intensity measurements of antibacterial treatment effect with 1D10 on *A. baumannii* NCTC13423::*luxCDABE* growth in the pig skin burn wound model.** At t = 5 h, the explants were treated with either 0, 25, 50, or 100 µg 1D10 (n = 3). Immediately after treatment with 1D10, the measured luminescence drops due to instant cell lysis inherent to the mode-of-action of engineered lysins [28]. This is visualized in the enlarged panel of the figure. The error bands indicate one standard deviation from the mean.

**Figure 5 pharmaceutics-14-02553-f005:**
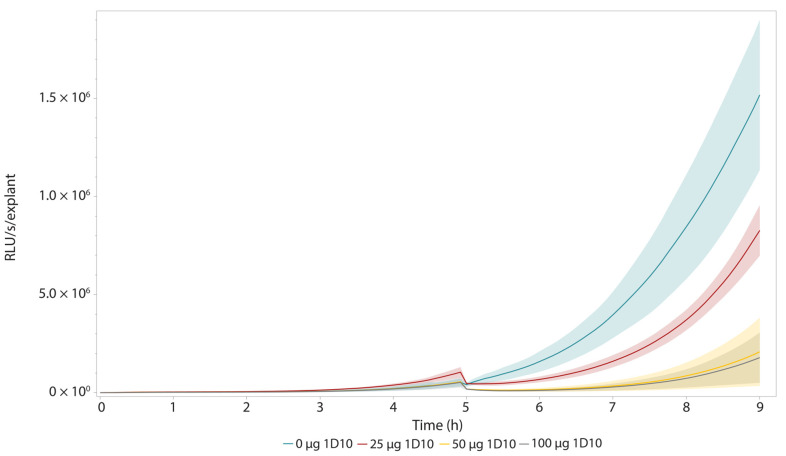
**Visualization of exponential regrowth in the first four hours after lysin treatment.** At t = 5 h, the explants were treated with either 0, 25, 50, or 100 µg of 1D10 (n = 3). Immediately after treatment with 1D10, the measured luminescence drops due to instant cell lysis inherent to the mode-of-action of engineered lysins [28]. After this drop, growth of the surviving cells retakes but at a slower pace in comparison with the untreated explant. This growth rate was quantified by fitting exponential curves and calculating the derivatives of these fitted curves, listed in Table 2. The error bands indicate one standard deviation from the mean.

**Table 1 pharmaceutics-14-02553-t001:** **Characterization of the antibacterial effect of the engineered lysin 1D10**. The minimum values upon antibacterial treatment were determined together with the time required to obtain these minima. The final columns display the calculated area under the curve for the treatments in the first four hours after treatment, the time frame of interest upon administration of this fast-acting enzybiotic. The values represent the average of three biological replicates (n = 3) and the standard deviation. Both the drop in bioluminescence intensity and the areas under the curves corresponding to the antibacterial effect were compared with the control using a two-tailed Student’s *t*-test. For the comparison of the time needed to reach the relative minimum bioluminescence intensity measurement, a one-sided *t*-test was performed. These data are graphically illustrated in Appendix A.

Treatment	Relative Drop in Luminescence (%)	*p*-Value	Time to Reach Minimum (h)	*p*-Value	Area under the Curve (t5–t9) (×102)	*p*-Value
0 µg 1D10	20.3 ± 9.31	N/A	0.0833 ± 0	N/A	41.0 ± 4.41	N/A
25 µg 1D10	58.3 ± 17.0	0.0405	0.278 ± 0.210	0.125	10.0 ± 3.87	0.000858
50 µg 1D10	79.4 ±8.41	0.00129	0.694 ± 0.337	0.0440	4.29 ± 3.47	0.000346
100 µg 1D10	77.0 ± 16.4	0.0121	0.639 ± 0.315	0.0464	4.97 ± 3.69	0.000481

**Table 2 pharmaceutics-14-02553-t002:** **Coefficients of fitted exponential curve between 5 h and 9 h post-infection.** The exponential curves have the syntax f(x) = a e^bx^. Additionally, the R^2^ value of the fit and the derivative of the fitted functions are represented in the table.

Treatment	a	b	f’(x)	f’(9)-f’(5)	R^2^ of Fit
0 µg 1D10	950	0.845	803 e^0.845x^	1.56 × 10^6^	0.980
25 µg 1D10	638	0.794	507 e^0.794x^	6.17 × 10^5^	1.000
50 µg 1D10	215	0.745	160 e^0.745x^	1.24 × 10^5^	0.990
100 µg 1D10	150	0.770	116 e^0.770x^	1.13 × 10^5^	0.991

## Data Availability

Not applicable.

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
