# Peer review of "A Bioluminescence-Based Ex Vivo Burn Wound Model for Real-Time Assessment of Novel Phage-Inspired Enzybiotics"

_pharmaceutics, 2022, doi:10.3390/pharmaceutics14122553_

Round 1
Reviewer 1 Report
This is a good proof of concept manuscript. Elegantly done.
Please address the following:
1. Are the authors implying to generate a systematic plasmid collection with different Abx selection marker and fluorescently tagged? It would be very powerful.
2. 3D tissue model that enables visualisation of efficiency of biologics is appreciated. Please can the authors consider figures that reflect this? For example, in cancer spheroid drug resistant studies, images of bioluminescence-labelled transfection are presented.
3. As with any eukaryotic cell experiments, are the explant synchronised for subsequent experiment. It's noted the authors mentioned "normalised" in the next paragraph (line 133). Please explain how the triplicates are set up vs controls. Are these biological replicates or technical replicates? Noted in lines 213, 246, 286, biological replicates was indicted. Maybe put this in the materials and method section.
4. Sharing of such plasmids are appreciated. Please describe (maybe in the supplementary, Illumina sequencing/protocol used to verify pBGlux and pBGlux_CmR plasmids respectively.
5. 5 hours post infection and data collection points - are these arbitrary criteria or is there a biological reasoning? For example, in 3D cancer spheroids experiment, drug resistance is measured based on half life of the biologics, cell proliferation and transfection rates. Would this apply to bacteriophage-derived engineered lysin on pig skin?
6. Setting up ex vivo models for translational research is appreciated. Please can authors consider looking into the pig skin interaction to the presence of say 1D10, ie was there an increase of proinflammation genes from the ex vivo model - is that a direct correlation to the biofilm as well as 1D10? What will the ex vivo model's interaction be in the presence of polymicrobial infections? How would synthetic lysin (precision) interact with each of these pathogens that have specialised colony on the pig skin?
Minor comment: Please use proper nomenclature for enzymes. e.g., first three letters are italicised.
Reviewer 2 Report
This is a well written manuscript and highlights good application of bioluminescence tagging of bacteria for assessment of novel antibacterial agents in an infected ex-vivo model. Whilst the majority of the results were fine, I am not convinced of the interpretation of the biofilm biomass quantification differences between the tagged and untagged A.Baumannii in the explants. Perhaps the authors could define the statistical differences they found as certainly the three replicates demonstrate clear differences.
Having said that, this is an interesting concept and would be useful for the readership.
